# Resilience in Older People: A Concept Analysis

**DOI:** 10.3390/healthcare11182491

**Published:** 2023-09-08

**Authors:** Gabriella Santos Lima, Ana Laura Galhardo Figueira, Emília Campos de Carvalho, Luciana Kusumota, Sílvia Caldeira

**Affiliations:** 1School of Nursing of Ribeirão Preto, University of São Paulo, Ribeirão Preto 14040-902, Brazil; ana.figueira@usp.br (A.L.G.F.); ecdcava@usp.br (E.C.d.C.); kusumota@eerp.usp.br (L.K.); 2Centre for Interdisciplinary Research in Health, Institute of Health Sciences, Universidade Católica Portuguesa, 1649-023 Lisbon, Portugal

**Keywords:** aged, concept formation, geriatric nursing, resilience psychological, standardized nursing terminology

## Abstract

(1) Background: Resilience has been presented as a potential protective factor to be promoted in difficult experiences in older people. However, further clarification of the concept of resilience for this population is required, as this is of critical interest for nursing care. (2) Aim: To develop the concept of resilience in older people to establish the elements that refer to the nursing outcome. Personal resilience (1309) from the Nursing Outcomes Classification (NOC), specifically in older people. (3) Methods: Concept analysis using Beth Rodgers’ evolutionary model. The attributes, antecedents, consequents, and empirical elements were described in the integrative review, with searches in PubMed, CINAHL, PsycINFO, LILACS, and Embase databases. A total of 2431 citations have been identified, and 110 studies were included. (4) Results: The concept of “resilience in older people” is composed of two attributes, available resources and positive behaviors, and is defined as positive attitudes of older people with the assistance of resources available from experiences of adversity. Conclusion: This analysis and concept development of resilience in older people provided sensitive indicators for nursing care in the context of adversity, considering available resources and with positive attitudes during this phase of life span.

## 1. Introduction

In its evolution, resilience was initially established as an innate characteristic, a personal attribute experienced by individuals who succumbed to major fatalities [1]. In current models, it is understood as a human potential to be developed during life, considering personal contexts, mechanisms, and abilities learned [2]. Self-control, problem-solving, coping, determination, and wisdom are abilities in the face of life experiences and especially present in favorable contexts with robust social, family, and spiritual structures that reinforce resilience [3,4].

In health, understanding the concept of resilience is critical due to its role in coping and adaptation in the face of adversity [3]. Different stages of life span lead to the development of resilience through critical experiences, such as illness, disability, living with violence, prejudice, and wars [4,5]. The more resilient, the more adaptive outcomes and less impact on health, considering the biological, psychological, and social aspects [3,6].

Nursing recognized worldwide for its role in health care, is responsible for evaluating and intervening in health promotion, disease prevention, treatment, rehabilitation, and recovery actions. In these contexts, nurses are trained to develop care plans based on the assessment of global and individualized needs, which mainly serve vulnerable groups in society [7].

As for 2021 global demographics, older people represent 13.5% (1 billion) [8] of the population. Interestingly, the dynamic character of resilience is expressed by more or less adaptive attitudes facing the complexity of experiences in advanced age [9]. Keeping active, having a purpose in life, managing physical and cognitive limitations, and performing self-care activities, among other adaptation and recovery behaviors, all suggest healthy functioning in older people [4,9].

However, research [9,10] highlights the essential role of social support and protection networks, harmonious family relationships, access to health services, and social coexistence that allow real conditions of assistance to older people’s demands. The consonance of protection mechanisms, life experiences, and internal and external resources is configured to promote longevity by supporting the challenges experienced and the limitations imposed by the senescent process [9,10]. From the perspective of traditional models of human aging, as a continuum, under the influence of environmental, social, individual and lifestyle determinants [8], resilience is remarkable, as it helps older people adapt and recover from vulnerable conditions [11]. Promoting resilience while considering the heterogeneity of the context of older people may drive the advancement of nursing care, particularly of the care for this population.

The study of the concepts in nursing corresponds to a greater understanding of the phenomena related to care. The attributes, antecedents, consequents, and empirical elements fundamentally contribute to nurses’ clinical reasoning [12]. Specifically, for the standardized language systems concerning nursing taxonomies, the theoretical aspects and conceptual structures are critical in developing a common language and nursing communication [13]. As such, the development of concepts relates to improving nursing knowledge, which is not static but evolutionary, as the characteristics, such as resilience, differ over time. This concept has been explored from an ageing perspective. However, in nursing, taxonomies keep incipient, especially in older people.

The purpose of this concept analysis is to develop the concept of resilience in older people to establish the elements that refer to the nursing outcome Personal resilience (1309) from the Nursing Outcomes Classification (NOC) [14], specifically in older people, based on the identification of the attributes, the antecedents, the consequences, and the empirical elements found in the literature.

## 2. Materials and Methods

The evolutionary model by Beth Rodgers [15] of concept analysis was used to develop “resilience in older people”. Six steps determine the process: (1) identification of the concept of interest and associated terms; (2) selection of the appropriate field for data collection; (3) data collection and identification of the attributes of the concept, considering the interdisciplinary, sociocultural context and temporal variations (antecedent and consequent); (4) analysis of the concepts’ characteristics; (5) identification of an example of the concept (if appropriate); (6) identification of implications, hypotheses for future development of the concept [12]. This concept analysis was based on the procedures described for conducting an integrative review [16].

### 2.1. Identification of the Concept of Interest and Associated Terms

In this study, the concept of resilience related to ageing is developed under the health care approach for a sensitive outcome to nursing interventions. Thus, the concept of interest was defined as “resilience in older people”, and the associated terms were:Psychological resiliencePsychological adaptationCoping

### 2.2. Data Sources

We selected five databases: PubMed (National Library of Medicine National Institutes of Health) from Bethesda, MD, USA; CINAHL (Cumulative Index to Nursing and Allied Health Literature) from Ipswich, MA, USA; PsycINFO (American Psychological Association) from Washington, USA, considering only PsycArticles; LILACS (Latin American Caribbean Literature on Health Sciences) from São Paulo, Brazil, and Embase^®^, Amsterdam, The Netherlands, with their controlled descriptors Medical Subject Headings (MeSH), Health Science Descriptors (DeCS), Entree Thesaurus and uncontrolled (keywords) combined using the Boolean operators “AND” and “OR” (Table 1).

### 2.3. Data Collection and Identification of Attributes, Antecedents and Consequents of the Concept

To achieve the objectives, the identification of the empirical elements proposing to evaluate the concept in the context of older people was also considered.

Initially, the guiding questions were defined using one of the variations of the acronym PICo (Population: people over 65 years of age; Interest: resilience; Context: ageing). The guiding questions were: What are the conceptual elements of resilience in older people? What are the measures that allow the assessment of resilience in older people?

As inclusion criteria, primary (experimental, observational, methodological, and analytical) and reflective studies, whose authors studied resilient characteristics in older people; research published electronically in Portuguese, English or Spanish; studies using samples of people aged 65 and over in any life situation or health condition were considered. No publications with time restrictions were considered.

The study selection process (Figure 1) was based on the PRISMA (Preferred Reporting Items for Systematic Reviews and Meta-Analysis) [17]. The study citations selected in the databases were imported to the software EndnoteWeb^®^ (access by: https://www.myendnoteweb.com/EndNoteWeb.html, accessed on 1 August 2023) [18] to remove duplicates. Then, Rayyan^®^ (access by: https://rayyan.ai/, accessed on 1 August 2023) [19] was used to organize the readings of titles, abstracts, and full text and to screen the studies independently among the reviewers.

In August 2020, a total of 1988 studies were found in the databases, and 202 were removed for duplicates. According to the inclusion criteria, after evaluating the titles and abstracts, 117 studies were selected for full-text reading; 46 studies were excluded (21 did not mention the concept or measure of resilience; nine did not report older people; six were literature reviews; two were written in other languages; two were congresses’ abstracts; two were dissertations; one was an editorial; one was an experimental study with animals; one was a concept analysis; one was unavailable for reading). After selection, 71 studies were included.

The literature analysis took more time than planned, aligned with the design of the concept of “resilience in older people”, and the search was updated in December 2022. From August 2020 to December 2022, 443 studies were added, considering the five databases. Sixty-one duplicate studies identified by EndnoteWeb^®^ [18] and Rayyan^®^ [19] were removed. According to the eligibility criteria, 332 studies were excluded after titles and abstracts evaluation, and 11 studies were excluded in full-text reading, as eight focused on the variable correlated of resilience and three for not having concept and resilience assessment. In this update, 39 more studies were included; therefore, for concept development, the total was 110 (Figure 1).

### 2.4. Data Analysis

For the analysis and synthesis of the included studies, a spreadsheet was created in Microsoft Office Excel^®^, containing information regarding the identification of the study (title, authors, year, country, and journal), the descriptors/keywords, the objective of the study, methodological aspects (design, sample characteristics, and study setting), and concept characteristics (substitute/related terms; attributes/definitions; antecedents; consequents; empirical elements).

Considering one of the purposes of the concept development, the attributes, antecedents, consequents, and empirical elements established the elements that make up the nursing outcome in the taxonomy of Nursing Outcomes Classification (NOC) [14]. The correspondence [20] of the elements of the concept analysis with the elements of the taxonomy was considered as follows: the attributes corresponded to the definition and title of the nursing outcome, the antecedents, and consequents to the indicators of the nursing outcome, and the empirical elements to the measurement scale (Figure 2).

## 3. Results

### 3.1. Characteristics of the Studies

A total of 110 were included (Appendix A).

Most were carried out in the United States of America *(n* = 37), Canada (*n* = 12), China (*n* = 10), and Brazil (*n* = 9), published between 1990–2022, with the majority (*n* = 39) between the years 2020–2022. There was a predominance of publications using a quantitative design (*n* = 66), as well as qualitative studies (*n* = 28), cohorts (*n* = 17), methodological studies (*n* = 8), and studies with an approach to older people in the context of life in the community (*n* = 38), in the presence of depressive symptoms (*n* = 5), under limitations, such as the risk for falls, visual impairment, bone fracture and diagnoses of Dementia and Stroke (*n* = 5), and in other situations of adversity, such as immigrants, widowhood, palliative care, holocaust, and others. After updating the database search, there were also studies reporting the resilience of older people in the context of the COVID-19 Pandemic (*n* = 8). This section may be divided by subheadings. It should provide a concise and precise description of the experimental results, their interpretation, and the experimental conclusions that can be drawn.

### 3.2. Concept Elements

#### 3.2.1. Definition

Different definitions of resilience were found and analyzed according to attributes, namely: Ecological construct [21,22,23,24,25,26,27,28,29,30]; Dynamic process of the lifecycle [31,32,33,34,35,36,37,38,39,40]; Protective factors [41,42,43,44,45,46,47] Positive perspective [31,35,40,42,48,49,50,51,52,53,54]; Multidimensional [40,42,55,56,57,58,59,60,61]; Adaptation [51,56,62,63,64,65]; Psychological resources [54,66,67,68]; Ageing [40,50,69,70,71,72]; Coping [48,53,73,74]; Self-management [33,75]; Innate/Hereditary [76].

#### 3.2.2. Surrogate and Related Terms

During the study analysis, other substitutes and related terms were identified, which may express similar ideas and characteristics, using different terminologies to represent the concept of interest [15].

With the perspective of coping associated with resilience, the terms “resilient coping patterns”, “coping resilience”, “resilient coping”, “adaptive coping” [31,42,48,65,73,77,78,79]; for ageing and resilience, “resilient ageing” and “resilience elderly” [69,80]; and for psychology and resilience, “psychological resilience”, “emotional resilience”, and “cognitive resilience” [43,66,70,72,81]. Other terms were also found, such as “resilience protective factors” [21], “resilience factors” [82], “physical resilience” [72,83,84,85], “biological resilience” [84], “health resilience” [72], “resilience of quality of life” [71], “resilient reintegration” [72], “spiritual resilience” [44], “cultural resilience” [55] “community resilience” [55], “social-ecological resilience” [86], “family resilience” [62,87], “sustainability” [88], “individual and interpersonal resilience” [62] “resilient identity” [11], “dispositional resilience” [72], “resiliency” [32], “multimorbidity resilience” [57,57,59,61], and “dietary resilience” [89].

#### 3.2.3. Attributes

The attributes constitute the real definition of the concept, which can be represented by situations or words that adequately characterize the concept [15].

As attributes of the “resilience in older people”, the literature identified resources available and positive behaviors (set of attitudes). Resources available [27,44,45,48,57,62,64,72,85,90] is characterized in studies as (environmental capital; cultural capital; economic capital; social capital; social relationships; family and friends; love and friendship; having divine support; faith in God; spirituality) and positive behaviors (set of attitudes) [11,24,27,34,42,49,52,53,57,64,66,67,70,72,91] as (courage and strength; resistance; hardiness; positive sense of self and an optimistic outlook on life; strong positivity demonstrated by identity projects, redemptive sequences and narrative openness; positive comparison with others; interpersonal control; mastery; belief in self; competence; maintaining purpose; determination; sense of purpose; self-determination; strives toward goal achievement; prosocial behavior; expressing gratitude; sense of humor; ability to use humor; problem-solving skills; meaningful work and activities; flexibility; creativity; autonomy; recovery; sustainability; warrior; adaptive; moving-on; self-esteem; esteem; strong self-efficacy).

#### 3.2.4. Antecedents

A total of eight antecedents were found in the literature. They are sociodemographic characteristics [11,21,22,23,24,25,26,27,28,29,31,32,33,34,35,36,37,38,39,41,42,43,44,45,46,48,49,50,51,52,53,54,55,56,57,58,59,60,62,63,64,66,67,69,70,71,72,73,75,76,77,78,79,80,81,82,83,84,85,86,88,89,90,91,92,93,94,95,96,97,98,99,100,101,102,103,104,105,106,107,108,109,110,111,112,113,114,115,116,117,118,119,120,121,122,123,124,125,126,127]; adversities experiences [11,21,22,34,42,45,46,48,54,58,63,67,69,71,83,91,111,113,115,119,120,123,125]; life experiences [33,42,48,58,69,81,84,98,120]; physiology factors [68,84]; social context (personal relationships and environmental support) [21,22,26,27,29,30,34,36,37,42,49,51,55,71,99,100,107,108,116,119,124,127,128]; intrinsic aspects (have purpose or goals; maintenance balance; beliefs) [37,42,45,49,51,53,58,60,62,64,67,77,82,98,107,116,117,119,120]; health conditions (lifestyle and express emotions) [21,30,35,45,50,53,56,57,60,62,72,74,91,100,108,111,112,113,116,124,125] and express self-awareness [33,37,42,45,47,48,58,60,77,98,119]. They characterize the antecedents of a concept, events or happenings that occur before the phenomenon, including the relationships with its temporal and sociocultural context [15].

#### 3.2.5. Consequents

Described as results, manifestations, or outcomes of the concept or phenomenon of interest [15], the consequences of resilience in older people, according to the literature, comprise the components of mental health (control of psychological symptoms and control of emotions) [11,22,23,38,40,43,50,53,54,59,67,68,69,79,80,82,88,95,96,100,126]; positive perspective and experience of ageing (active aging; independence and autonomy; values socialization; personal behaior) [11,21,22,25,26,29,31,34,39,40,44,45,47,48,49,50,60,61,63,64,69,70,73,75,78,81,82,85,86,88,95,97,100,101,103,104,106,107,108,109,111,112,116,121,127,128,129]; grief and loss experience [42,43,51,72,114]; coping strategies [26,31,40,44,64,69,73,89,91,98,104,105,109,123]; health perspective [27,41,46,56,72,79,84,114,118]; optimistic perspective [39,45,51,60,62,76,81,86,96,108,116].

#### 3.2.6. Empirical Elements

The empirical elements represent categories or classes related to the essential attributes in which the concept can be observed and that allow its operational definition [15]. Some authors proposed to measure the concept of resilience in the context of life of older people, using scales and/or interventions and qualitative assessment.

Among the scales, the Connor Davidson Resilience Scale (Four domains—grit, active coping self-efficacy, accommodative coping self-efficacy e spirituality) [35,36,40,41,46,54,56,68,76,80,81,90,95,97,100,104,108,110,116,123,124,129]; The Resilience Scale (Two domains—personal competence; acceptance of self and life) [33,49,51,58,66,75,78,83,103,106,116,125]; Simplified Resilience Score = Leave Behind Questionnaire + Resilience Scale [43,105,118]; Brief Resilient Coping Scale [35,44,49,56,79,91,102,110,123,130]; Dispositional Resilience Scale [36,53,67]; Groningen Ageing Resilience Inventory [42]; The Hardy-Gill Resilience Scale [23]; Resilience in Older Adults Survey [57,88]; The Ego-Resilience Scale [67]; John Henry Active Coping level [79]; Psychological Resilience Scale for Adults [24,70]; Psychological Resilience Against Physical Difficulties Index [117].

For studies with a qualitative approach to resilience, it was observed, in the objectives of the evaluations, the interest in exploring significant life events and experiences [21,32,39,48,52,60,77,119]; exploring events of adversity and confrontation [29,34,47,60]; exploring resilience [26,28,44,45,111]; exploring support mechanisms [39,47,63,77,82]; measuring social support [22,25,28,77,127]; measuring self-efficacy [22]; measuring self-awareness [30,40,112]; physical and cognitive performance [51,87]; exploring what is getting old [39,127]; resilient characteristics [39,86,127]. Specifically, in the context of the COVID-19 Pandemic, the questions focused on exploring challenges experienced [25]; how the COVID-19 Pandemic affected their lives [25]; what the changes in that period were; social networks and assistance from government agencies or communities during the pandemic [25]; serious financial impact during the COVID-19 Pandemic; strategies, resources and processes do older adults [86].

#### 3.2.7. Exemplar Case

To demonstrate the application and purposes of clarifying the concept of “resilience in older people” [15], an exemplary case was identified in the literature inserted in the context of older people’s lives [131].

“…Mrs. W is a 75-year-old woman, widowed, who lives alone in her apartment after her husband died three years ago. She was admitted to a community hospital with chief complaints of falling at home and pain in her left knee. Her three children live in different states, and she has a female friend who is a neighbor in her apartment building, and they talk on the telephone at least twice a week. She has a degree in Music and enjoys playing the piano at her church. She relies on her strong faith in God and on the support from her pastor and church. She does her house chores and grocery shopping and enjoys cooking. Her main concerns are falling again, her painful left knee, and being able to resume her walking regimen. Mrs. W said, “I am going to do whatever I can to help myself get back to normal again…” [131].

In this context, regarding the episode of Mrs. W’s fall, the resilience attributes identified were family, friends, church, faith, belief in God, having an apartment (resources available) and “playing the piano”; “prosocial behavior” (positive behaviors). As antecedents: pain, fall, husband died (adversity); and consequents: “I am going to do whatever I can to help myself get back to normal again…” (optimistic perspective); “and being able to resume her walking regimen” (positive perspective and experience of ageing—active ageing); “grocery shopping and enjoys cooking” (positive perspective and experience of ageing—independence and autonomy).

Figure 3 represents the concept of resilience in older people, integrating the attributes of the concept and its etiology (antecedents), outcomes (consequents) and empirical elements, as well as the conceptual definition of “positive attitudes of older people with the assistance of resources available from experiences of adversity”.

## 4. Discussion

Clarifying the concept of resilience constitutes the development of mechanisms to help older people face the challenges of ageing. The concept of “resilience in older people” was related to the presence of available resources and behaviors that reflect positive attitudes during the experiences of adversity in ageing. Thus, it is proposed as a conceptual definition of “positive attitudes of older people with the assistance of resources available from experiences of adversity”.

Access to resources from social, economic, cultural, family, environmental and spiritual aspects is configured as a necessary interface to limitations, weaknesses, and a decrease in the vital capacity of older people [132]. In a global context, we can witness the expressive increase in longevity through the integrated guarantee of these resources, which support survival and provide conditions for active ageing with good quality of life [133,134].

For older people, greater resilience in situations of vulnerability is related to daily life, and behaviors such as self-efficacy, use of good humor, problem-solving skills, and interpersonal control, among others, are critical in strengthening personal perspectives of ageing [135]. Life experiences promote global development and attitudes that facilitate protecting available resources and increase resilient mechanisms related to adaptation and coping [134].

Studies analyzing the concept of resilience are associated with cardiovascular disease [130], dementia [136], mortality [137], nurses [138] and genetics [139] in overall literature about nursing care. Establishing the concept, defining its temporal characteristics, and exploring motivations and outcomes support nurses’ clinical reasoning for more effective care interventions [140].

The etiologic events and characteristics, described as conceptual antecedents, were mediated by older people: higher education and income and having a partner [11,21,22,23,23,24,25,26,27,28,29,31,32,33,34,35,36,37,38,39,41,42,43,44,45,46,48,49,50,51,52,53,54,55,56,57,58,59,60,62,63,64,66,67,69,70,71,72,73,75,76,77,78,79,80,81,82,83,84,85,86,88,89,90,91,92,93,94,95,96,97,98,99,100,101,102,103,103,104,105,106,107,108,109,110,111,112,113,114,115,116,117,118,119,120,121,122,123,124,125,126,127]; having experienced adversities (impairment of living conditions, health issues, mental health and social challenges, exposure to trauma and prejudice) [11,21,22,34,42,45,46,48,54,58,63,67,69,71,83,91,111,113,115,119,120,123,125]; life experiences (satisfaction, reminiscences, wisdom and problem-solving practice) [33,42,48,58,69,81,84,98,120]; physiological factors (involving the functions of the autonomic nervous system, interactions of genetic, environmental, molecular and immune system causes) [68,84]; the social context (participates in family, social, religious relationships, among friends and neighbors; access to social security, community and health services (interacts with art and leisure) [21,22,26,27,29,30,34,36,37,42,49,51,55,71,99,100,107,108,116,119,124,127,128]; intrinsic aspects (aims at moving on with life; expresses perseverance, personal control and equanimity; has spiritual and religious support) [37,42,45,49,51,53,58,60,62,64,67,77,82,107,116,117,119,120]; health conditions (lifestyle that involves self-care activities, self-preservation, self-efficacy, independence and autonomy; express emotions through communication, humor, hope, and self-esteem) [21,30,35,45,50,53,56,57,60,62,72,74,91,100,108,111,112,113,116,124,125]; expresses self-awareness (demonstrates personal competences of acceptance, adaptation, identity, and self-reflection; search for self-improvement) [33,37,42,45,47,48,58,60,77,98,119].

Adversities were also described as antecedents of resilience in the context of family caregivers of older people with dementia. In different situations, adversities provoke reactions to deal with and overcome this problem. Family conflicts, caregiving tasks, role reversals, stigma, social stress, gender role identity, and rigid expectations were necessary precursors to the development of caregivers’ resilient potential [92].

For the consequents of “resilience in older people”: mental health in older people (it presents control of depressive symptoms, apathy and anxiety; control of negative emotions and stress; expresses positive emotions, stability and emotional maturity) [11,22,23,38,40,43,50,53,54,59,67,68,69,79,80,82,88,95,96,100,126]; positive perspectives of aging (expresses quality of life and satisfaction, wisdom in coping with vulnerabilities; seeks to remain active; develops strategies for autonomy and independence; social engagement) [11,21,22,25,26,29,31,34,39,40,44,45,47,48,49,50,60,61,63,64,69,70,73,75,78,81,82,85,86,88,95,97,100,101,103,104,106,107,108,109,111,112,116,121,127,128,129]; experiences of grief and loss (demonstrates recovery and continues active after the loss; does not express denial) [42,43,51,72,114] coping strategies (avoids stressful situations, demonstrates problem-solving skills, develops strategies to alleviate adversity and expresses courage and performance) [26,31,40,44,64,69,73,89,91,98,104,105,109,123]; health perspectives (positive self-report health; has adequate physical and mental health conditions; demonstrates less impact on illness) [27,41,46,56,72,79,84,114,118]; optimistic attitude (demonstrates a positive attitude, gratitude for life, and optimism) [39,45,51,60,62,76,81,86,96,108,116].

Another concept analysis, published in 2013 [93], also addressed the concept of “resilient ageing” and highlighted quality of life. “Resilient ageing” tend to demonstrate mainly life satisfaction and personal satisfaction, also suggested by the optimistic perspective and the positive perspective of aging in this concept analysis.

Regarding the operationalization of nursing care, evaluating human responses in clinical, family, and community situations is essential for clinical reasoning and, later in the stages of the nursing process, to establish care planning, interventions, and outcomes assessment [13,14,94].

The empirical elements of “resilience in older people” are available in a quantitative approach (scales, inventories, and surveys) and a qualitative approach (interviews, dialogues, and guiding questions). In both, the characteristics presented in Figure 3 are evaluated, such as coping [31,35,36,40,41,46,54,56,65,68,76,80,81,90,95,97,100,104,108,110,116,122,123,124,129], self-efficacy [31,35,36,40,41,46,54,56,65,68,76,80,81,90,95,97,100,104,108,110,116,122,123,124,129], personal competences [33,43,60,63,66,74,75,83,120,126], acceptance [33,66,74,75,83,120,126], life experiences [22,32,34,48,60,77,80] and adversities [26,34,48].

In the search update, eight studies were included with the theme of resilience in the context of older people during the pandemic. Situations of greater risk of complications from SARS-CoV-2 virus infection, social isolation, economic risk, imminent grief and family distancing aggravated the vulnerable scenario of the elderly [25,35,50,59,86,89,105,129]. Identified, through the resilient dynamics, the presence of attitudes to maintain control in the face of adversity [35], self-efficacy in measures to protect against infection [59,89,129], optimism [50] and the use of care and social resources [25,86,105] in older people show outcomes of greater adaptation during the pandemic.

Included in the NOC taxonomy 2008, the nursing outcomes Personal resilience (1309) [11] have had no review to this date and the main indicators are related to childhood and adolescence. This concept analysis proposes, with the conceptual elements and the definition, to develop the phenomenon of resilience from the specific perspective of older people.

Presenting in vulnerable contexts, the disposition for resilience thrives on adaptive coping with positive attitudes. It considers it among the models and nursing classifications that propose systematized care based on the expertise and clinical reasoning of nurses.

The aim and review question, which considered all the conceptual and empirical elements of the concept of “resilience in older people”, allowed the inclusion of many indexed studies in the databases. Thus, the limitation of this study was the extensive process of data screening and extraction from the included studies considering the rigor of the independent review and discussion of the elements that substantiated the concept.

## 5. Conclusions

The concept of “resilience in older people” was defined as the positive attitudes of older people with the assistance of resources available from experiences of adversity. Exploring conceptual elements, predisposing factors, and their manifestations in specific populations and conditions brings greater accuracy in nursing care. Particularly, the NOC relates to constructing and refining specific and measurable indicators, which are sensitive to nursing interventions and concretely represent the status, behavior, or perception. As so, this concept analysis has provided a definition and specific indicators of a new nursing outcome, resilience in older people, which is important for nursing care.

## Figures and Tables

**Figure 1 healthcare-11-02491-f001:**
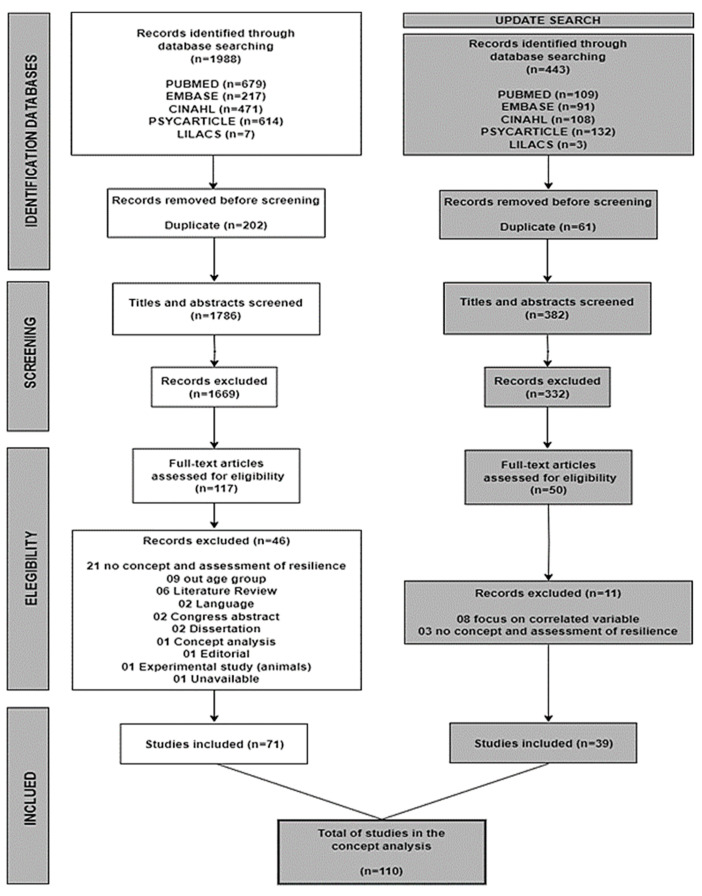
PRISMA flowchart of the study selection process.

**Figure 2 healthcare-11-02491-f002:**
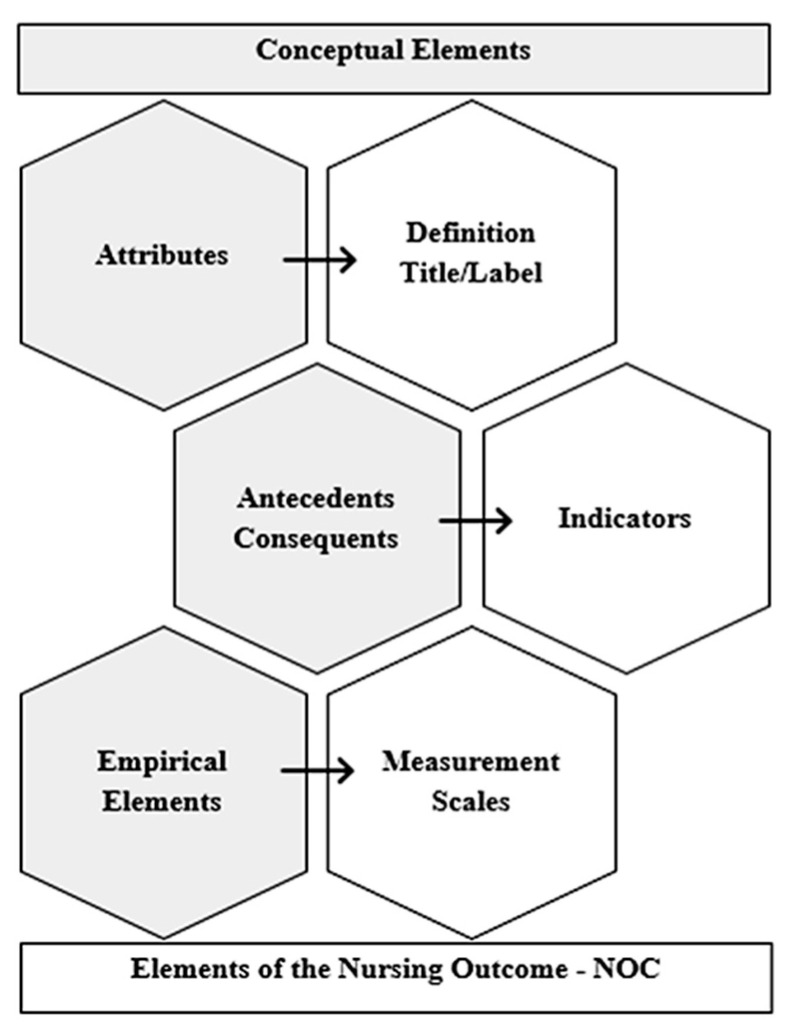
Correspondence between the elements of the concept analysis and Nursing Outcomes Classification (Authors’ work).

**Figure 3 healthcare-11-02491-f003:**
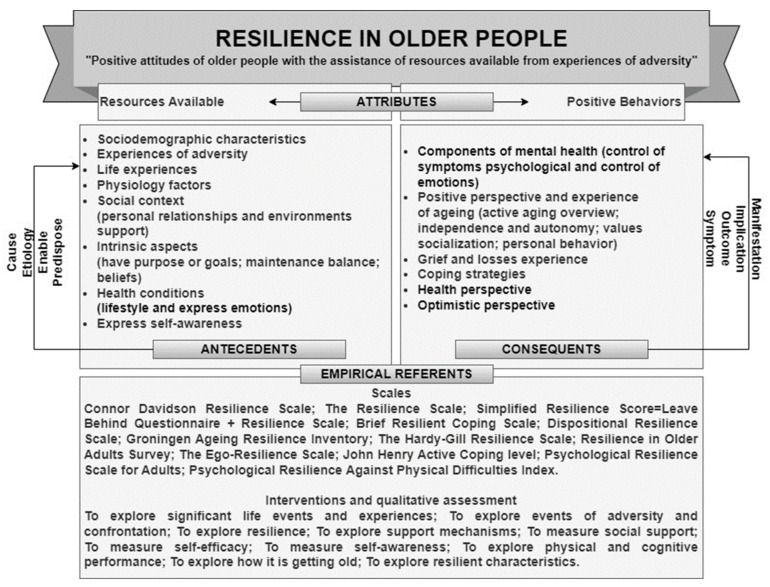
Concept of “Resilience in older people”.

**Table 1 healthcare-11-02491-t001:** Databases search strategy.

Databases	Search Strategy
Pubmed	aged [MeSH Terms] OR aged 80 and over [MeSH Terms] OR older people OR oldest old AND resilience psychological [MeSH Terms] OR resilience OR resiliency AND adaptation psychological [MeSH Terms] OR adaptive capacity OR coping AND aging [MeSH Terms] OR late life
Embase	aged OR very elderly OR elderly OR aged 80 and over OR very old AND psychological resilience OR resilience OR resiliency AND coping behavior OR coping OR adaptation psychological OR psychologic adaptation AND Aging
Cinahl	MH aged OR MH aged 80 and over OR MH older people OR oldest old AND MH hardiness OR psychological resilience OR resilience OR resiliency AND MH psychological adaptation OR adaptive capacity OR MH coping AND MH aging OR late life
PsycInfo (Psycarticles)	Index Terms: aged OR aged, 80 and over OR elderly OR older people OR oldest old AND Index Terms: resilience psychological OR resilience OR resiliency AND Index Terms: coping behavior OR coping OR adaptive capacity OR adaptation psychological AND Index Terms: aging OR late life
LILACS	mh: idoso OR aged OR anciano OR idoso de 80 anos ou mais OR aged, 80 and over OR anciano de 80 o más años OR pessoa idosa OR pessoa de idade OR elderly OR persona de edad OR persona mayor AND mh: resiliência psicológica OR resilience psychological OR resiliencia psicológica OR resiliencia OR resiliente OR resilience OR resiliency AND adaptação psicológica OR adaptation, psychological OR adaptación psicológica OR enfrentamento OR coping AND mh: envelhecimento OR aging OR envejecimiento

Search in August 2020 and updated in December 2022.

## Data Availability

Not applicable.

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
