# Peer review of "Resilience in Older People: A Concept Analysis"

_healthcare, 2023, doi:10.3390/healthcare11182491_

Round 1

Reviewer 1 Report

Thanks for giving me the opportunity to reviewing this paper.

This compelling paper employs concept analysis to elucidate the notion of "resilience in the elderly"—a significant area of research. However, before considering it for publication, several issues require attention. Kindly refer to my comments below:

Introduction:

1. Enhance clarity and coherence:

A. The initial sentences about resilience's role could benefit from some restructuring to flow better.

B. Use consistent terminology. For instance, if you refer to "elderly" in one place, try to keep using that term throughout unless there's a need to differentiate.

2. Clarify or expand on specific ideas:

A. The concept of "biopsychosocial health" could be unfamiliar to some readers. Consider briefly defining or expanding on it.

B. Clarify "configured as the main support strategy for longevity." Does this mean that resilience is the main support strategy for longevity?

3. Grammatical and stylistic corrections:

A. "The elderly represents" should be "The elderly represent."

B. "the elderly" is a somewhat outdated and potentially non-inclusive term. Consider using "older adults" or "senior individuals" as alternatives.

C. Consider using a more active voice for certain sentences to make the text more engaging. For instance, "The development of resilience in different stages..." could be "Different stages of the lifespan develop resilience through..."

4. Sentence-by-sentence suggestions:

A. Original:

The role of resilience, coping and adaptation in the face of adversity makes critical the understanding of the concept of resilience in health [1].

Suggestion:

Understanding the concept of resilience in health is critical due to its role in coping and adaptation in the face of adversity [1].

B. Original:

Likewise, life experiences in favorable contexts and with adequate social, family, cultural and spiritual structure strengthen the skills of self-control, problem-solving, coping, determination and wisdom [2,4].

Suggestion:

Life experiences, especially in favorable contexts with robust social, family, cultural, and spiritual structures, bolster skills such as self-control, problem-solving, coping, determination, and wisdom [2,4].

5. Add transitions for smoother reading:

A. Consider adding transitional phrases to better connect sections, like "In relation to global demographics," before mentioning the percentage of the elderly population in 2021.

6. Clarify the conclusion:

A. The closing remarks about the study's purpose are clear, but consider leading with this purpose earlier on to set the stage for your discussion, allowing readers to understand the context from the beginning.

Materials and Methods

1. Paragraph 2:

A. Replace "Six steps determine it;" with "The process is determined by six steps:"

It might be helpful to use a colon instead of a semicolon before listing the six steps.

2. Paragraph 2.1:

A. Clarify the phrase "resilience focusing on ageing" to "resilience as it relates to ageing."

Consider using bullet points for the terms associated with the concept for easier readability.

3. Paragraph 2.2:

A. The sentence "This concept analysis was based on the procedures as described for conducting an integrative review [13]." seems out of place. Consider moving it to the beginning of the paragraph.

"In total, five databases were selected:" can be more concise as "We selected five databases:"

4. Table 1:

A. Consider using a clear format for your search terms. For example, maintain consistent usage of quotes and brackets around MeSH terms and keywords.

B. For clarity, break up search strings with line breaks or indentation to indicate separate sets of terms combined with AND/OR.

5. Paragraph 2.2:

A. The sentence "This concept analysis was based on the procedures as described for conducting an integrative review [13]." seems out of place. Consider moving it to the beginning of the paragraph.

B. "In total, five databases were selected:" can be more concise as "We selected five databases:"

Results

1. Clarity and Readability

A. What is the classification-based line for AX, BX, CX, EX, HX, etc.? It is necessary to clarify.

B. Use a more consistent naming or reference system for the studies. The current alphanumeric codes (like A1, B2, etc.) are hard to follow and can make the passage less intuitive.

Discussion

The "Discussion" section is akin to an extension of the results. The author should compare their findings with previous studies and provide specific recommendations for future research.

Some sentences need revision for better fluency.

Reviewer 2 Report

I read your article with great interest. However, I am concerned about the following points, which led me to this opinion.

Major points

The World Health Organization's definition of the older people is 65 years and older. However, the countries presented in the results (Table S1) do not necessarily define 65 as older people. This point needs to be clarified before discussing the results.

The search terms in this REVIEW include "aged, 80 and over" and "oldest old". Can I consider this age group as elderly all together? For example, there is a large difference in physical and cognitive functions between 65 and 85 years old, and I believe that bias cannot be denied in the results obtained. 

Too few mentions of study limitation. Please clarify any biases inferred from the methods and results.

Minor points

Words such as "elderly" and "older people" are mixed. Unification is needed.

Words like "COVID-19" and "SARS-CoV-2 virus infection" are mixed up. Unification is needed.

Author Response

Please see the attachment."

Round 2

Reviewer 1 Report

The authors have addressed all of my concerns. I have no further comments.

Reviewer 2 Report

Thank you for your careful revisions.

I have the impression that it is easier to read than last time.

I hope that it will be read by many readers and cited in the paper.